# Antioxidant and Metal-Chelating Activities of Bioactive Peptides from Ovotransferrin Produced by Enzyme Combinations

**Hiruni Sashikala Wickramasinghe [1], Edirisinghe Dewage Nalaka Sandun Abeyrathne [1,2], Ki-Chang Nam [2] and Dong Uk Ahn [3,*]**

1. Department of Animal Science, Uva Wellassa University, Badulla 90000, Sri Lanka
2. Department of Animal Science & Technology, Sunchon National University, Suncheon 57922, Korea
3. Department of Animal Science, Iowa State University, Ames, IA 50011, USA
* Correspondence: duahn@iastate.edu

**Abstract:** Peptides produced from food sources possess numerous bioactivities that make them useful in improving human health and preventing diseases. Although many studies related to egg protein hydrolysis are available, little work has been conducted on the production of bioactive peptides from apo-ovotransferrin (OTF) using two-step enzyme hydrolysis. The objectives of this study were to produce bioactive peptides from OTF using two enzymes and to determine their functional properties. Lipolyzed OTF was prepared at a concentration of 20 mg/mL and treated with protease (3 h at 55 °C), papain (3 h at 37 °C), elastase (24 h at 25 °C), and α-chymotrypsin (3 h at 37 °C) as the first enzyme treatment. The hydrolysates from the first step of hydrolysis were treated with the above enzymes in different combinations and incubated for 24 h at their optimum temperatures, followed by heat inactivation at the end of each treatment. Based on 15% SDS-PAGE results, the nine best enzyme combinations were selected for further analysis. Papain + protease (PapPro, 0.0075 ± 0.004 malondialdehyde (MDA) mg/kg), α-chymotrypsin + papain (ChyPap, 0.081 ± 0.003 MDA mg/kg), and elastase + α-chymotrypsin (ElaChy, 0.083 ± 0.015 MDA mg/kg) showed strong antioxidant activity. PapPro showed the highest Fe-chelating activity (5.40 ± 0.85%) but lacked Cu-chelating activity. In conclusion, PapPro, ChyPap, and ElaChy treatments of OTF produced peptides with strong antioxidant and Fe-chelating activities but lacked Cu-chelating activity. Thus, ovotransferrin hydrolysates produced using PapPro, ChyPap, and ElaChy treatments have the potential to reduce oxidative stress in the body.

**Keywords:** ovotransferrin; hydrolysates; enzyme combinations; antioxidant activity; metal-chelating activity

## 1. Introduction

Bioactive peptides are defined as specific protein fragments that positively affect body functions and health [1]. They play important roles in the metabolic functions of living organisms and display numerous bioactivities [2]. Recently, there has been an increasing demand for functional foods because scientific evidence has shown that natural bioactive peptides in foods have beneficial effects in improving human health and preventing diseases [3,4]. Bioactive peptides have been used as food ingredients to prevent the oxidation and microbial degradation of foods during processing and storage [1,5]. The sources of bioactive peptides are milk, meat, eggs, fish, and plants, such as maize, soy, rice, mushroom, and pumpkin [1]. Eggs are a protein-dense food and a rich source of bioactive proteins with a low economic cost [5]. Peptides derived from egg proteins display diverse biological activities, including antihypertensive, antioxidant, antimicrobial, anti-inflammatory, antidiabetic, and Fe-/Ca-binding activities [2].

Enzymatic hydrolysis is the most widely used approach for producing bioactive peptides from egg proteins [6]. Egg-white proteins have significant biological properties that can be useful in the nutraceutical and pharmaceutical industries [7]. Many studies have focused on the production of bioactive peptides from egg-white proteins. However, little work has been conducted on the production of bioactive peptides from ovotransferrin using two-enzyme combinations. Ovotransferrin is the second major protein in egg whites and accounts for 12–13% of the total egg-white proteins. Ovotransferrin has a single glycopeptide chain containing 686 amino acids and 15 disulfide bonds, with a molecular weight of 76 kDa. This single chain consists of two homologous N and C lobes, further divided into two similarly sized sub-domains [8,9].

The hydrolysis of a protein by different enzymes, enzyme combinations, and incubation conditions diversifies the size, function, and amino acid sequence of the peptides produced [2]. Bioactive peptides from the major egg-white proteins, ovalbumin, ovomucoid, and ovomucin, have been produced using proteases, papains, elastases, trypsin, and $\alpha$-chymotrypsin [10–12]. Effective functional peptides produced from egg-white proteins are usually smaller than 2 kDa in molecular size [13]. Most enzymes used to hydrolyze proteins have unique cleavage sites; thus, hydrolyzing proteins with two enzymes will be more effective than a single-enzyme treatment in producing peptides with small molecular weights and characteristics [14].

Whole hydrolysates from ovotransferrin with unique functional properties have immense potential for commercial use in functional foods as a main component [15]. This would be highly desirable because it can directly add antioxidant and antimicrobial functions and nutritional value. It increases the quality and shelf life of foods while assuring consumer safety and adding beneficial effects to human health [3]. Ovotransferrin hydrolysates also have great potential for use in the nutraceutical and pharmaceutical industries [7,16]. Hence, the production of bioactive peptides from ovotransferrin using different enzyme combinations would increase the value and use of eggs.

The objective of this study was to produce bioactive peptides using combinations of two enzymes, including protease, papain, elastase, and $\alpha$-chymotrypsin, and to determine the functional properties of their hydrolysates.

## 2. Materials and Methods

### 2.1. Materials

Apo-ovotransferrin, prepared according to a method described in a previous study [9], was obtained from the Iowa State University, Ames, USA. Protease (Pro) from *Bacillus licheniformis* (Alcalase® 2.4 L; $\geq$2.4 U/g solutions; P4860), papain (Pap) from papaya latex ($\geq$10 U/mg protein; P4762), elastase (Ela) from porcine pancreas ($\geq$4.0 U/mg protein, E1250), and $\alpha$-chymotrypsin (Chy) from bovine pancreas ($\geq$40 U/mg protein; C4129) were purchased from Sigma-Aldrich (St. Louis, MO, USA). Other chemicals were purchased from Sigma-Aldrich (St. Louis, MO, USA), Daejung Chemical and Materials (Gyeonggi-do, Korea), and Research Lab Fine Chemical Industries (Mumbai, India).

### 2.2. Enzymatic Hydrolysis of Ovotransferrin

Ovotransferrin was hydrolyzed using two enzymes in sequence: one enzyme was hydrolyzed after the other. The results of single-enzyme hydrolysis in a previous study [17] were used to determine the best temperature and time combinations for the first step of hydrolysis. Ovotransferrin solution (20 mg/mL) was prepared using distilled water, and the pH was adjusted to optimize the conditions for each enzyme (Pro, pH 6.5; Pap, pH 6.5; Chy, pH 7.6; and Ela, pH 7.8) at room temperature. An enzyme–substrate ratio of 1:100 was used, and the incubation temperature and time were: Pro, 3 h at 55 °C; Pap, 3 h at 37 °C; Chy, 3 h at 37 °C; and Ela, 24 h at 25 °C. After incubation, samples were heated at 100 °C for 15 min in a water bath. The resulting solutions were centrifuged at $2000\times g$ for 20 min at 4 °C (ST 40R; Thermo Fisher Scientific, Waltham, MA, USA), and the supernatant was collected and subjected to a second hydrolysis step after adjusting the pH of the

hydrolyzed solution. The optimum pH and temperature of each of the second enzymes were determined for 0, 3, 6, 9, 12, and 24 h in a water bath (YCW-010E; Gemmy Industrial Corp., Taipei, Taiwan), followed by heat inactivation at 100 °C for 15 min. The samples were centrifuged using a micro-centrifuge (Centrifuge 5427R; Eppendorf AG, Hamburg, Germany), and the resulting supernatant was freeze-dried using a freeze dryer (FD 5512; ShinBioBase Co., Ltd., Dongducheon, Korea) and used as the hydrolysate. The degree of hydrolysis was analyzed using 15% SDS-PAGE.

### 2.3. Antioxidant Activity Assessment Using the 2-Thiobarbituric Acid-Reactive Substance Assay

Antioxidant activity was measured according to the method described in a previous study [10], with some modifications. An oil-in-water emulsion was prepared by homogenizing 1 g of pure refined soybean oil (Sajo, Korea), 100 µL of Tween-20, and 100 mL of distilled water using a polytron homogenizer (D-500; Scilogex, Rocky Hill, NJ, USA) for 2 min at full power. Then, 8 mL of the oil emulsion, 1 mL of distilled water, and 1 mL of ovotransferrin hydrolysate were mixed and incubated at 37 °C for 16 h. After incubation, 1 mL of the incubated sample was mixed with 2 mL of TBA/TCA (Thiobarbituric acid/Trichloroacetic acid) (20 mM TBA/15% TCA ($w/v$)) solution, and 50 µL of 10% butylated hydroxyanisole in 90% ethanol was added and vortexed. The mixture was incubated in a 90 °C water bath for 15 min and then centrifuged at $3000 \times g$ for 15 min. The absorbance of the solution was measured using a UV–visible spectrophotometer (J.P. Selecta, Barcelona, Spain) at 532 nm against a blank prepared with 1 mL of distilled water and 2 mL of TBA/TCA solution. The malondialdehyde (MDA) level was calculated using the standard curve and expressed as milligrams of MDA per liter (MDA mg/L) of the emulsion.

### 2.4. $Fe^{2+}$-Chelating Activity

The Fe-chelating activity of the hydrolysate was measured using the ferrozine method described in a previous study [18]. The hydrolysate (100 µL) was transferred to a 15 mL Falcon test tube, mixed with 0.9 mL of distilled water and 1 mL of 10 ppm $Fe^{2+}$ ($FeSO_4$), and incubated for 5 min at room temperature (25 °C). The mixture was added to 900 µL of 11.3% TCA and centrifuged at $2500 \times g$ for 10 min. One milliliter of the supernatant was transferred into a disposable culture tube, along with 1 mL of distilled water, 800 µL of 10% ammonium acetate, and 200 µL of ferroin color indicator, and then vortexed. After 5 min of incubation at room temperature, absorbance was measured at 562 nm. The Fe-chelating activity was measured using the following equation:

$$\text{Fe chelating activity } (\%) = 1 - \left( \frac{\text{Absorbance of the sample}}{\text{Absorbance of the blank}} \right) \times 100 \tag{1}$$

### 2.5. $Cu^{2+}$-Chelating Activity

The Cu-chelating activity of hydrolyzed ovotransferrin was measured using a method described in a previous study [19], with some modifications. One milliliter of hydrolysate and 1 mL of 0.2 mM $CuSO_4$ were mixed in a 15 mL Falcon tube and incubated for 5 min at room temperature. Then, 1 mL of 11.3% TCA solution was mixed with the sample, followed by centrifugation at $2500 \times g$ for 10 min. Then, 2 mL of the supernatant was transferred to a disposable culture tube, mixed with 1 mL of 10% pyridine and 20 µL of 0.1% pyrocatechol violet (Sigma-Aldrich), and incubated for 5 min at room temperature. Samples were centrifuged at $2500 \times g$ for 10 min, the absorbance of the supernatant was measured at 632 nm, and the Cu-chelating activity was measured using the following equation:

$$\text{Cu chelating activity } (\%) = 1 - \left( \frac{\text{Absorbance of the sample}}{\text{Absorbance of the blank}} \right) \times 100 \tag{2}$$

*2.6. Statistical Analysis*

All tests were replicated thrice, and the data were analyzed using Minitab 19. One-way ANOVA in a completely randomized design was used to analyze the bioactivities of the hydrolysates. Data are expressed as the mean $\pm$ standard deviation.

## 3. Results and Discussion

*3.1. Hydrolysis of Ovotransferrin*

Enzymatic hydrolysis is one of the most used methods for producing bioactive peptides from precursor proteins [20]. Based on the catalytic properties of substrate specificity, different oligopeptides can be obtained via enzyme–substrate interactions [21]. Therefore, various enzymes have been used to produce bioactive peptides from egg-white proteins. In previous studies, different enzymes, such as Pramod 278P, thermolysin [14], trypsin, pepsin [2], neutrase, alcalase, protamex, protex, flavorzyme, $\alpha$-chymotrypsin, maxazyme, collupulin [22], thermolysin, and pepsin [23], have been tested for the hydrolysis of ovotransferrin.

In this study, ovotransferrin was treated with Pro (3 h at 55 °C), Pap (3 h at 37 °C), Ela (24 h at 37 °C), or Chy (3 h at 37 °C) as the first enzyme treatment. The hydrolysates of the first enzyme were treated with the second enzyme using combinations of the enzymes used in the first-step hydrolysis, which generated 12 enzyme combinations that were then incubated for 0–24 h at their optimal temperatures. Peptides with molecular weights lower than 2 kDa were not retained in the 15% SDS-PAGE gel [10]. As the degree of hydrolysis increases, the density of the bands decreases, and eventually, no bands are visible [14]. Therefore, the gel images were used to determine the degree of hydrolysis. According to the results, nine treatments were selected as the best to hydrolyze ovotransferrin. The enzyme combinations and conditions selected were as follows:

Pro + Pap (3 h at 37 °C, ProPap), Pro + Chy (3 h at 37 °C, ProChy), Chy + Pro (3 h at 55 °C, ChyPro), Chy + Pap (3 h at 37 °C, ChyPap), Ela + Chy (3 h at 37 °C, ElaChy), Ela + Pap (3 h at 37 °C, ElaPap), Ela + Pro (3 h at 55 °C, ElaPro), Pap + Pro (3 h at 55 °C, PapPro), and Pap + Chy (3 h at 37 °C, PapChy) (Figures 1–3). These nine treatments were used to analyze their functional properties, such as antioxidant and metal-chelating activities.

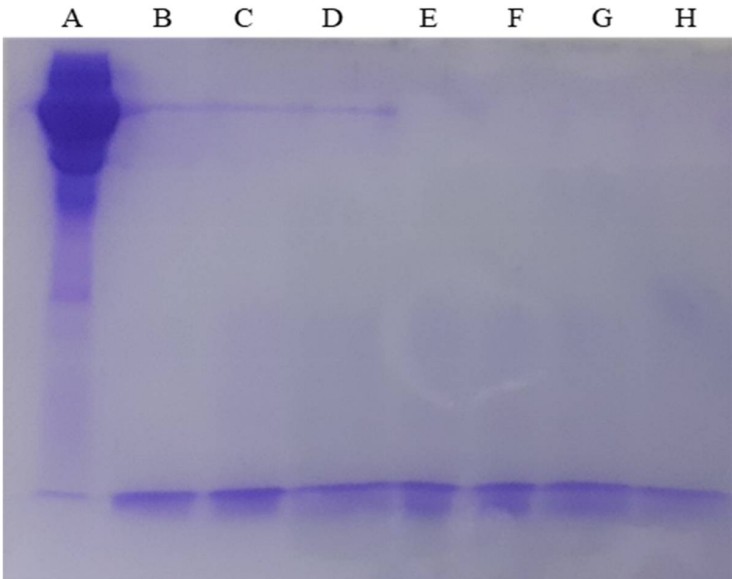

**Figure 1.** Results of 15% SDS-PAGE of ovotransferrin hydrolyzed with protease and papain. Lane A = ovotransferrin (OT); Lane B = OT hydrolyzed with protease for 3 h at 55 °C; Lanes C to H = OT hydrolyzed with protease followed by papain at 65 °C for 0, 3, 6, 9, 12, and 24 h (purified ovotransferrin was used as the marker to compare the level of hydrolysis).

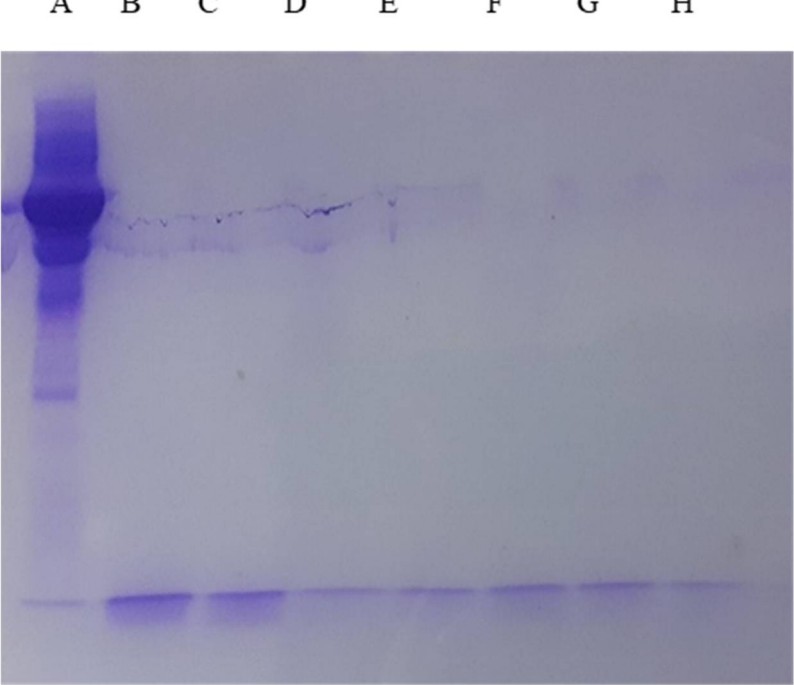

**Figure 2.** Results of 5% SDS-PAGE of ovotransferrin hydrolyzed with protease and α-chymotrypsin. Lane A = ovotransferrin (OT); Lane B = OT hydrolyzed with protease for 3 h at 55 °C; Lanes C to H = OT hydrolyzed with protease followed by α-chymotrypsin at 50 °C for 0, 3, 6, 9, 12 and 24 h (purified ovotransferrin was used as the marker to compare the level of hydrolysis).

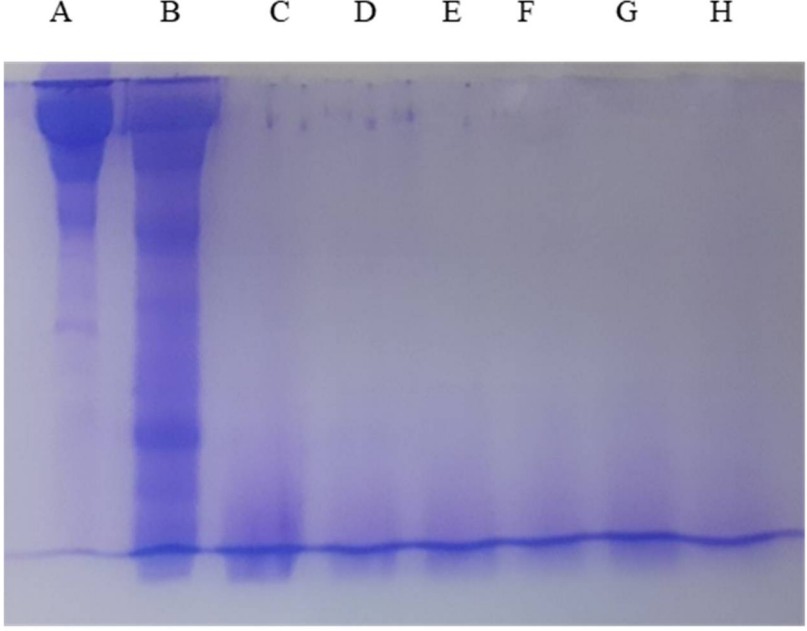

**Figure 3.** Results of 15% SDS-PAGE of ovotransferrin hydrolyzed with elastase and papain. Lane A = ovotransferrin (OT); Lane B = OT hydrolyzed with elastase for 24 h at 37 °C; Lanes C to H = OT hydrolyzed with elastase followed by papain at 65 °C for 0, 3, 6, 9, 12, and 24 h (purified ovotransferrin was used as the marker to compare the level of hydrolysis).

*3.2. Antioxidant Activity of the Hydrolysates*

Due to increasing health concerns, consumers prefer naturally derived foods. The identification and development of natural antioxidants from plant and animal sources are

important [22]. Numerous studies have focused on the production of bioactive peptides with antioxidant properties from food proteins. Recent evidence has shown that bioactive peptides from egg proteins may be a good source of antioxidants [24].

Ovotransferrin, the second major egg-white protein, has also been reported to have strong antioxidant activity. MDA is an important oxidation end-product and is considered the main marker of lipid peroxidation. Owing to the reactivity of TBA with several reactive substances, thiobarbituric acid-reactive substances are commonly used as indicators of oxidative stress in biological systems [25]. As shown in Figure 4, the oil emulsion with ovotransferrin or its hydrolysates had significantly lower thiobarbituric acid-reactive substance values than those of the control. Among the tested combinations, PapPro (0.0075 ± 0.004 MDA mg/kg), ChyPap (0.081 ± 0.003 MDA mg/kg), and ElaChy (0.083 ± 0.015 MDA mg/kg) treatments showed the highest antioxidant properties in the oil emulsion ($p < 0.05$). The hydrolysis of proteins with different enzymes alters their antioxidant potential. This may be partly due to the action of enzymes that cleave peptide bonds at specific amino acids and produce a sequence of amino acids with different antioxidant effects [22]. Therefore, the hydrolysis of polypeptides in various regions may result in hydrolysates with varied radical-scavenging activities. The results of the present study suggest that specific peptides with potent antioxidant activity can be generated by hydrolyzing ovotransferrin using the three enzyme combinations mentioned above. However, further research is needed to obtain more conclusive information on the exact peptide sequences, sizes, and amino acid compositions for the antioxidant potential and other functions of the peptides produced.

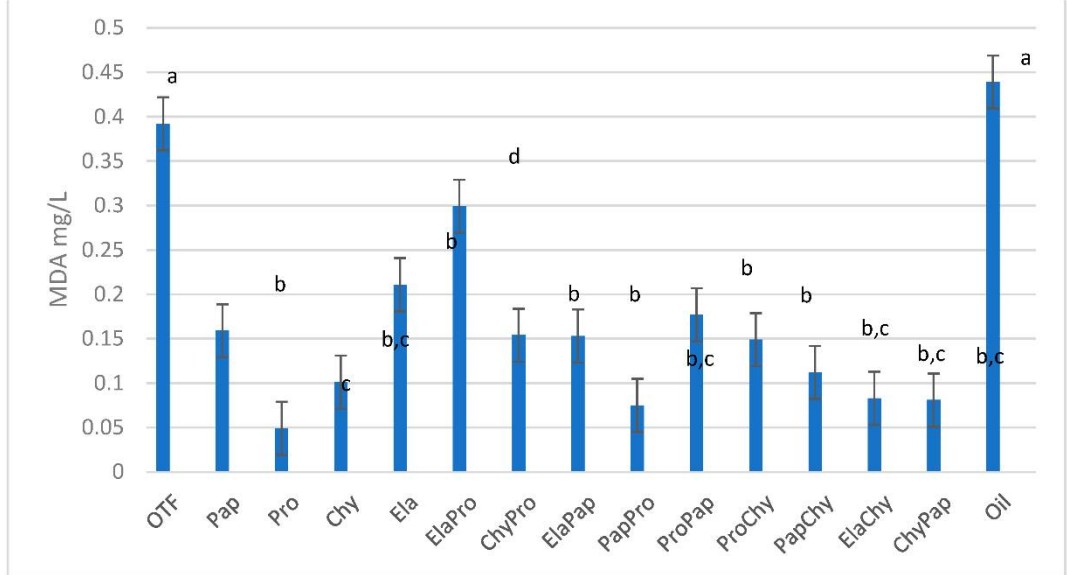

**Figure 4.** Graphical expression of TBA-reactive substance value of oil emulsion (mg of malondi-aldehyde/L) of the hydrolysates from ovotransferrin hydrolyzed with two-enzyme combinations with a comparison with ovotransferrin and single-enzyme treatments [OTF = ovotransferrin (OT), Pap = OT hydrolyzed with papain (3 h at 37 °C), Pro = OT hydrolyzed with protease (3 h at 55 °C), Chy = OT hydrolyzed with α-chymotrypsin (3 h at 37 °C), Ela = OT hydrolyzed with elastase (24 h at 25 °C), ElaPro = OT hydrolyzed with elastase + protease (3 h at 55 °C), ChyPro = OT hydrolyzed with α-chymotrypsin + protease (3 h at 55 °C), ElaPap = OT hydrolyzed with elastase + papain (3 h at 37 °C), PapPro = OT hydrolyzed with papain + protease (3 h at 55 °C), ProPap = OT hydrolyzed with protease + papain (3 h at 37 °C), ProChy = OT hydrolyzed with protease + α-chymotrypsin (3 h at 50 °C), PapChy = OT hydrolyzed papain + α-chymotrypsin (3 h at 37 °C), ElaChy = OT hydrolyzed with protease + α-chymotrypsin (3 h at 37 °C), ChyPap = OT hydrolyzed with α-chymotrypsin + papain (3 h at 37 °C)]. [a–d] Values with different letters indicate significant differences between the treatments ($p < 0.05$).

### 3.3. Metal-Chelating Activity of the Hydrolysates

Ovotransferrin belongs to the transferrin family [26]. The global structures of ovotransferrin and transferrin are identical but differ only in their attached glycan chain and pI [27]. Ovotransferrin shows 50% homology with mammalian transferrin and lactoferrin [28]. The significant structural similarities between ovotransferrin and transferrin explain the similarity in their biological roles, such as transporting Fe from the plasma to cells or regulating Fe levels in biological fluids [15].

As shown in Figure 5, enzymatic hydrolysis significantly increased the $Fe^{2+}$-chelating activity of ovotransferrin, but some of the hydrolysates showed $Fe^{2+}$-releasing activity. The highest $Fe^{2+}$ chelation among the treatments was observed with ovotransferrin treated with PapPro ($5.40 \pm 0.85$) ($p < 0.05$). However, ovotransferrin treated with ProPap and ChyPro showed opposite reactions. In addition to Fe transport to target cells, transferrin has been proposed to be involved in many other cellular events. Previous studies have reported that metal-binding peptides can reduce microbial growth and lipid oxidation [26]. The ovotransferrin molecule comprises two homologous halves, each containing a single Fe-binding site, and is well-known for its high affinity for Fe, which is associated with antibacterial properties [27]. Therefore, PapPro is effective for producing peptides with metal-chelating activity.

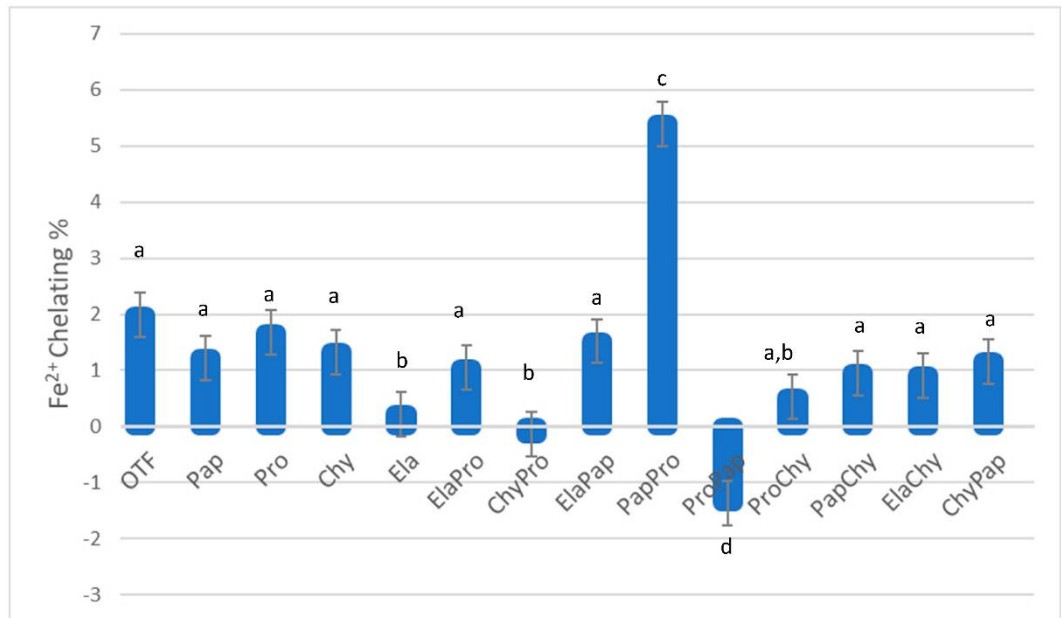

**Figure 5.** Graphical expression of $Fe^{2+}$-chelating activity of the hydrolysates from ovotransferrin hydrolyzed with two-enzyme combinations in comparison with ovotransferrin and single-enzyme treatments [OTF = ovotransferrin (OT), Pap = OT hydrolyzed with papain (3 h at 37 °C), Pro = OT hydrolyzed with protease (3 h at 55 °C), Chy = OT hydrolyzed with α-chymotrypsin (3 h at 37 °C), Ela = OT hydrolyzed with elastase (24 h at 25 °C), ElaPro = OT hydrolyzed with elastase + protease (3 h at 55 °C), ChyPro = OT hydrolyzed with α-chymotrypsin + protease (3 h at 55 °C), ElaPap = OT hydrolyzed with elastase + papain (3 h at 37 °C), PapPro = OT hydrolyzed with papain + protease (3 h at 55 °C), ProPap = OT hydrolyzed with protease + papain (3 h at 37 °C), ProChy = OT hydrolyzed with protease + α-chymotrypsin (3 h at 50 °C), PapChy = OT hydrolyzed papain + α-chymotrypsin (3 h at 37 °C), ElaChy = OT hydrolyzed with protease + α-chymotrypsin (3 h at 37 °C), ChyPap = OT hydrolyzed with α-chymotrypsin + papain (3 h at 37 °C)]. [a–d] Values with different letters indicate significant differences between the treatments ($p < 0.05$).

However, all of the hydrolysates showed negative Cu-chelating activity (Figure 6). The ovotransferrin protein showed a low level of Cu-chelating activity ($5.76 \pm 0.76\%$), but all of the hydrolysates showed Cu-releasing activities. Similar results were obtained for

hydrolysates produced from ovomucin using papain and protease enzymes. Therefore, ovotransferrin hydrolysates produced using the two-enzyme combinations are not suitable for Cu chelation.

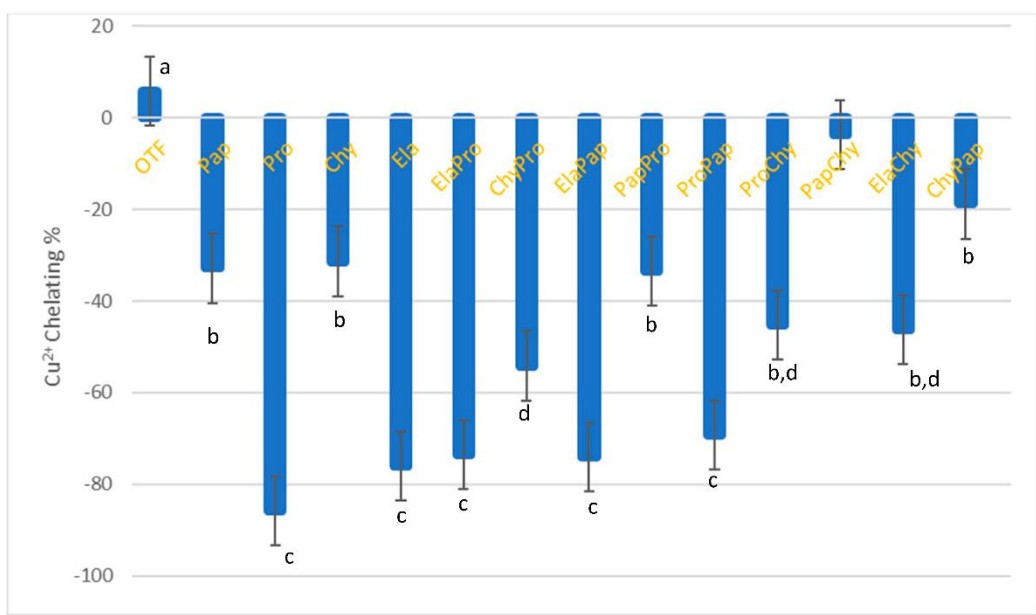

**Figure 6.** Graphical expression of $Cu^{2+}$-chelating activity of the hydrolysates from ovotransferrin hydrolyzed with two-enzyme combinations in comparison with ovotransferrin and single-enzyme treatments [OTF = ovotransferrin (OT), Pap = OT hydrolyzed with papain (3 h at 37 °C), Pro = OT hydrolyzed with protease (3 h at 55 °C), Chy = OT hydrolyzed with $\alpha$-chymotrypsin (3 h at 37 °C), Ela = OT hydrolyzed with elastase (24 h at 25 °C), ElaPro = OT hydrolyzed with elastase + protease (3 h at 55 °C), ChyPro = OT hydrolyzed with $\alpha$-chymotrypsin + protease (3 h at 55 °C), ElaPap = OT hydrolyzed with elastase + papain (3 h at 37 °C), PapPro = OT hydrolyzed with papain + protease (3 h at 55 °C), ProPap = OT hydrolyzed with protease + papain (3 h at 37 °C), ProChy = OT hydrolyzed with protease + $\alpha$-chymotrypsin (3 h at 50 °C), PapChy = OT hydrolyzed papain + $\alpha$-chymotrypsin (3 h at 37 °C), ElaChy = OT hydrolyzed with protease + $\alpha$-chymotrypsin (3 h at 37 °C), ChyPap = OT hydrolyzed with $\alpha$-chymotrypsin + papain (3 h at 37 °C)]. [a-d] Values with different letters indicate significant differences between the treatments ($p < 0.05$).

## 4. Conclusions

The ovotransferrin hydrolysates produced with Pap (3 h at 37 °C) followed by Pro (3 h at 55 °C) hydrolysis (PapPro); Chy (3 h at 37 °C) followed by Pap (3 h at 37 °C) hydrolysis (ChyPap); and Ela (24 h at 25 °C) followed by Chy (3 h at 37 °C) hydrolysis (ElaChy) showed excellent antioxidant and $Fe^{2+}$-chelating activities. Thus, ovotransferrin hydrolysates produced using PapPro, ChyPap, and ElaChy treatments have the potential to reduce oxidative stress in the body.

**Author Contributions:** Conceptualization, D.U.A., K.-C.N. and E.D.N.S.A.; methodology, D.U.A. and E.D.N.S.A.; software, H.S.W.; validation, D.U.A.; formal analysis, H.S.W.; investigation, H.S.W.; resources, D.U.A., K.-C.N. and E.D.N.S.A.; data curation, H.S.W.; writing—original draft preparation, H.S.W.; writing—review and editing, D.U.A., K.-C.N. and E.D.N.S.A.; visualization, D.U.A. and E.D.N.S.A.; supervision, D.U.A. and E.D.N.S.A.; project administration, D.U.A. All authors have read and agreed to the published version of the manuscript.

**Funding:** This research received no external funding.

**Institutional Review Board Statement:** Not applicable.

**Informed Consent Statement:** Not applicable.

**Data Availability Statement:** Not applicable.

**Conflicts of Interest:** The authors declare no conflict of interest.

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
