# Peer review of "Antioxidant and Metal-Chelating Activities of Bioactive Peptides from Ovotransferrin Produced by Enzyme Combinations"

_poultry, doi:10.3390/poultry1040019_

Round 1
Reviewer 1 Report
The authors used several enzymes in different combinations to hydrolyze ovotransferrin. The antioxidant and metal-chelating activity of the resultant hydrolysates were evaluated. The enzymolysis strategy is novel and interesting. I suggest this manuscript can be published after some modifications. Here are some comments to help improve the quality of this paper.
1. Line 82: the purity of ovotransferrin? Was the apo-form or holo-form ovotransferrin used for enzymolysis? Please include the information.
2. Line 96-97: was the enzyme: substrate ratio calculated according to the mass or enzyme activity?
3. Why SDS-PAGE was used to determine the degree of hydrolysis? There is a specific method (OPA) based on the content of free -NH2 groups.
4. Line 127-129: what is the temperature for the Fe2+ chelating experiment? Five minutes for incubation seems a little short. TCA was used to precipitate the hydrolysates, but have the authors considered the acid condition TCA provided which could disrupt the binding between Fe2+ and hydrolysates? The accurate results might not be reflected. Actually, ethanol may be more suitable for protein or peptide precipitation under this circumstance.
5. Line 138-139: the same question.
6. Fig.1-3: generally, Tricine SDS-PAGE is more suitable to characterize the molecular weight distribution of protein hydrolysates because some peptides with small molecular weight might exist. Also, no protein Marker bands are presented in Fig.1-3.
7. Line 168: How can the authors calculated the degree of hydrolysis through the profile the SDS-PAGE? The color of the bands were quantified here?
8. Fig.4-6 are not well-arranged, it is suggested to address them. As for the metal-chelating data, why negative values were presented? Accord to the equation, if no metal-chelating activity was shown, the values would be near zero. It seems confused to me.
Author Response
File loaded

Reviewer 2 Report
The work introduce bioactive OVT peptides with antioxidant and metal-chelating activities. However, I do not believe the manuscript achieve level of the journal because of the following points.
1. The OVT derived peptides have largely reported in previous literature. Although two-step enzyme hydrolysis has been processed, the innovation is still not enough. Thus, the experiments are suggested to be carefully redesigned.
2. The experiments were relatively ordinary. The sequences of peptides have not been detected. In the similar research, the peptide identification is necessary. Because we do not know whether there are some new peptides in the hydrolysate in this work.
3. The evaluation of peptide was too simple. There are many ways to evaluate antioxidant or other biological functions of peptides. More approaches and experiments are suggested to be supplemented to enrich the work.
4. More latest references should be referenced and cited in the manuscript.
Author Response
File loaded.

Reviewer 3 Report
The SDS page gels in Figures 1, 2, and 3 are missing a Molecular weight marker. This would help the reader better estimate molecular weights of proteins.
After the 2 enzyme treatment, there is only major band depicted in the gel. Is it possible to run a different gradient to separate the bands at low MW? Would this tell the reader anything? Does addition of the second enzyme hydrolyze remaining OT and/or hydrolyze the smaller MW fragments? Can we draw conclusions about the MW fragments and antioxidant activity?
In Figure 4, the Pro = OT hydrolyzed with protease looks like it has the highest antioxidant properties in the oil emulsion, even better than the combinations. Is this correct? There is no discussion of this in the results section.
Author Response
File loaded

Reviewer 4 Report
- References are not updated after 2018, except for 2 references in 2021 as self citations.
- Some abbriviation meanings are missed such as: Thiobarbituric acid (TBA) and Trichloroacedic acid (TCA).
- I suggest to make an index for Fe/Cu chelating to fina the optimal treatment for both Fe and Cu. I think a different conclusion will be obtained.
Author Response
File loaded
